# Fast Calculation of Computer Generated Holograms for 3D Photostimulation through Compressive-Sensing Gerchberg–Saxton Algorithm

**DOI:** 10.3390/mps2010002

**Published:** 2018-12-20

**Authors:** Paolo Pozzi, Laura Maddalena, Nicolò Ceffa, Oleg Soloviev, Gleb Vdovin, Elizabeth Carroll, Michel Verhaegen

**Affiliations:** 1Delft Center for Systems and Control, Delft University of Technology, Mekelweg 2, 2628 CD Delft, The Netherlands; o.a.soloviev@tudelft.nl (O.S.); g.v.vdovine@tudelft.nl (G.V.); m.verhaegen@tudelft.nl (M.V.); 2Department of Imaging Physics, Delft University of Technology, Lorentzweg 1, 2628 CJ Delft, The Netherlands; l.maddalena@tudelft.nl (L.M.); n.g.ceffa@tudelft.nl (N.C.); e.c.m.carroll@tudelft.nl (E.C.); 3Flexible Optical B.V., Polakweg 10-11, 2288 GG Rijswijk, The Netherlands

**Keywords:** optogenetics, spatial light modulators, computer generated holograms

## Abstract

The use of spatial light modulators to project computer generated holograms is a common strategy for optogenetic stimulation of multiple structures of interest within a three-dimensional volume. A common requirement when addressing multiple targets sparsely distributed in three dimensions is the generation of a points cloud, focusing excitation light in multiple diffraction-limited locations throughout the sample. Calculation of this type of holograms is most commonly performed with either the high-speed, low-performance random superposition algorithm, or the low-speed, high performance Gerchberg–Saxton algorithm. This paper presents a variation of the Gerchberg–Saxton algorithm that, by only performing iterations on a subset of the data, according to compressive sensing principles, is rendered significantly faster while maintaining high quality outputs. The algorithm is presented in high-efficiency and high-uniformity variants. All source code for the method implementation is available as Supplementary Materials and as open-source software. The method was tested computationally against existing algorithms, and the results were confirmed experimentally on a custom setup for in-vivo multiphoton optogenetics. The results clearly show that the proposed method can achieve computational speed performances close to the random superposition algorithm, while retaining the high performance of the Gerchberg–Saxton algorithm, with a minimal hologram quality loss.

## 1. Introduction

The development of optogenetics tools such as channelrhodopsin-2 (ChR2) [1] encompasses optical methods to efficiently stimulate neurons for investigating how neural circuit structure is related to function. In the optical manipulation of neural circuits, proper choice of the light delivery method is crucial. Ideally, the optical system should be capable of targeting a wide range of regions of interest (ROIs), from single synapses to 3D populations of neurons, and provide a time resolution on the order of millisecond along with micrometer spatial resolution [2]. Common techniques to fulfill these requirements can be classified into two main categories: point-scanning methods and parallel excitation methods [3]. The first approach consists of a tightly focused laser beam rastered across the ROIs typically using galvanometric mirrors or acousto-optic devices (AOD) to control the scan pattern. Scanning approaches are limited by the time needed to steer the laser beam: although the scan sequence can be randomized with digital scanners, only one point is ever stimulated at a time [4]. In fact, the ROIs in the neuron under investigation must be activated within a short temporal window to trigger sufficient depolarization to local action potential.

Alternatively, parallel excitation methods project an image into the sample, simultaneously illuminating all ROIs. Digital micromirror devices (the optically active chip in most digital light projectors) are capable of such image projection, making use of an array micrometer-sized mirrors (“pixels”) to reflect light in or out of the image to form a pattern. These devices modulate amplitude of the incident light, the total power depends on the pattern, and it can be inefficient to generate the relatively high light intensity especially for two-photon stimulation [5].

Phase-only liquid crystal on silicon spatial light modulators (LCoS-SLM) are commonly used in optogenetics for the generation of complex illumination patterns within the sample. A LCoS-SLM is a two-dimensional, pixelated device capable of modulating the phase of a monochromatic, polarized light beam, without affecting its intensity. When positioned in the pupil plane of a microscope, SLMs can be used to modulate the intensity distribution of light at the image plane. The phase modulation distribution programmed on the SLM to obtain a desired illumination pattern is called a computer-generated hologram (CGH).

Computer-generated holograms provide a high level of flexibility in the spatial shaping of excitation light in two and three dimensions. The three-dimensional distribution of excitation does not impose a limit on the temporal resolution, which is only dependent on the residence time $R_{t}$, the refresh rate of the SLM (30–60 Hz), and the kinetics of the optogenetics tool [6]. The calculation of CGH is a non-trivial problem and several algorithms are available. In particular, two main classes of algorithms are available: fast Fourier transform based algorithms and multi-spot algorithms.

While fast Fourier transform algorithms can generate extremely complex illumination patterns in very short times, a strict requirement is that illumination light should be focused either on a two-dimensional plane [7], or at most on a limited number of of two-dimensional planes at different depths [8].

On the other hand, multi-spot algorithms tend to be slower, but can generate holograms of diffraction limited spots arbitrarily distributed in the three-dimensional field of view of the optical system. These spots can be used to stimulate sub-cellullar regions in the neuron of interest [9]. Alternatively, to use a multi-spot hologram to illuminate ROIs of the size of neurons, the entire illumination pattern can be scanned in a spiral or raster pattern [10], or combined with temporal focusing excitation [11]. Several algorithms exist to calculate a multi-spot hologram, generally representing a tradeoff between computation speed and quality of the hologram. A good summary of the most common algorithms for the generation of multi-spot holograms can be found in [12].

The most common way to compute a multi-spot CGH is the Gerchberg–Saxton (GS) algorithm [13], an iterative procedure of alternating projections allowing for the computation of high quality CGHs in time frames compatible with experimental needs. While the computational cost scales linearly with the amount of generated spots, in a general experimental scenario, calculation of a CGH can take between a few seconds and a couple of minutes. In optogenetics experiments requiring frequent calculation of new stimulation patterns “on-the-fly”, the random superposition (RS) algorithm is sometimes preferred to GS algorithm [14], sacrificing hologram quality to achieve higher computation speeds.

In this paper, a modified version of the GS algorithm, based on compressive sensing of the pupil function, is presented. The compressive sensing Gerchberg–Saxton algorithm (CS-GS) can generate holograms of equal quality to the standard GS algorithm, but reduces the computational cost by more than one order of magnitude, reaching time scales similar to the RS algorithm.

## 2. Method Description

### 2.1. Random Superposition and Gerchberg–Saxton Algorithm

The CGH $\phi (x^{\prime },y^{\prime })$ generating a single spot at coordinates (x,y,z) at the sample can be easily computed through Fourier optics, as a combination of tip, tilt and defocus aberrations
(1)$$\phi \left(x^{\prime },y^{\prime }\right)=\frac{\pi z}{\lambda f^{2}}\left(x^{\prime 2}+y^{\prime 2}\right)+\frac{2\pi }{\lambda f}\left(xx^{\prime }+yy^{\prime }\right),$$
where *f* is the equivalent focal length of the optical system and λ is the wavelength.

In the GS algorithm, the CGH generating a set of *N* spots at coordinates $\left(x_{n},y_{n},z_{n}\right)$ is assumed to be the phase of the superposition of all the fields generating the individual spots
(2)$$\phi_{\mathrm{cgh}}\left(x^{\prime },y^{\prime }\right)=\arg\sum_{n=1}^{N}\frac{1}{N}e^{-i(\phi_{n}\left(x^{\prime },y^{\prime }\right)+\theta_{n})},$$
where $\theta_{n}$ are some constant phase terms. The only degrees of freedom that can be addressed from the algorithm to optimize the quality of the CGH are the *N* values of $\theta_{n}$.

It is to be noticed that, by simply solving Equation (Equation 2) for random values of $\theta_{n}$, a low quality CGH can be calculated, without any optimization on the constant phase terms $\theta_{n}$. This is known as the RS algorithm, which has computational cost scaling as M×N where *M* is the number of SLM pixels.

To compute optimal values of $\theta_{n}$, the GS algorithm proceeds by alternating projections. The initial values of the phases $\theta_{n}^{0}$ are chosen at random, and the initial CGH $\phi_{\mathrm{cgh}}^{0}$ is computed according to Equation (Equation 2).

At each *k*th iteration, the values of $\theta_{n}^{k}$ are estimated as
(3)$$\theta_{n}^{k}=\arg\int_{SLM}e^{-i(\phi_{\mathrm{cgh}}^{k-1}-\phi_{n})}dx^{\prime }dy^{\prime },$$
and $\phi_{\mathrm{cgh}}^{k}$ is computed again according to Equation (Equation 2).

Performing a GS algorithm for *K* iterations, while generating *N* spots, on an SLM with *M* pixels has a computational cost scaling as K×N×M. Empirically, a few tens of iterations are required for convergence of the algorithm.

### 2.2. Compressive Sensing Approach

To understand the principle of the CS-GS algorithm described in this paper, it must be noticed that Equation (Equation 2) can be rewritten as
(4)$$\vec{y}=A{\vec{x}}^{T},$$
where $\vec{y}$ is a *N*-dimensional vector representing the field values for each spot, *A* is a N×M-dimensional matrix in which each *n*th line contains the value of $\frac{1}{N}e^{-i\phi_{n}}$ at each *m*th pixel of the SLM, and $\vec{x}$ is a vector of coefficients in the form $x_{m}=e^{-i\phi_{\mathrm{cgh},m}}$ where $\phi_{\mathrm{cgh},m}$ is the value of the phase of the CGH at the *m*th pixel of the SLM.

Since generally M≫N, this is an under-determined linear system. While an exact solution to the system is provided by Equation (Equation 2), according to the principles of compressive sensing [15], and in particular equivalently to *k*-space compressed imaging applications [16], a satisfying solution can be obtained by only sampling a subset of cM randomly distributed pixels of the SLM, where $\frac{N}{M}<c<1$.

According to this assumption, the first K−1 iterations of the GS algorithm can be performed only considering a subset of randomly distributed pixels of the SLM, and only on the *K*th iteration the whole CGH should be computed according to Equation (Equation 2). With this algorithm, the computational cost scales as K×N×cM+N×M, which, for c×K≪1, can be approximated as just N×M, which is equivalent to the RS algorithm cost. Reasonable values of *K* are in the order of $10^{2}$.

### 2.3. Weighted Gerchberg–Saxton

The GS algorithm tends to optimize the efficiency of the CGH, but does not guarantee uniform intensity amongst the foci created. In applications in which it is crucial for the CGH to generate spots of equal intensity, a modified version of the GS algorithm, named weighted Gerchberg–Saxton (WGS) algorithm [12] can be implemented to improve uniformity, with a minor loss in efficiency. The WGS algorithm has the same computational cost of GS. In the WGS algorithm, *N* additional degrees of freedom are introduced, as intensity weights $\omega_{n}$ in the combination of the spots wavefronts. The initial value of the weights is equal for all spots, and set to $\omega_{n}^{0}=\frac{1}{N}$. Equation (Equation 2) is then rewritten as:(5)$$\phi_{\mathrm{cgh}}^{k}\left(x^{\prime },y^{\prime }\right)=\arg\sum_{n=1}^{N}\omega_{n}^{k}e^{-i(\phi_{n}+\theta_{n})}.$$

Computing the intensity of each spot as:(6)$$I_{n}^{k}=\left|\int_{SLM}e^{-i(\phi_{\mathrm{cgh}}^{k-1}-\phi_{n})}dx^{\prime }dy^{\prime }\right|$$
at each step, the weights are recalculated as:(7)$$\omega_{n}^{k}=\omega_{n}^{k-1}\frac{\langle I_{n}^{k}\rangle }{I_{n}^{k}}.$$

This approach cannot be implemented in a compressive sensing framework, as it quickly diverges for even slightly inaccurate calculations of $I_{n}$. However, significant improvements can be obtained by running the CS-GS algorithm for K-1 iterations, and only performing a single iteration of the WGS at the end when computing the final CGH. This, however, doubles the computation time compared to CS-GS when c×K≪1. For the rest of the paper, this approach is referred to as compressive sensing weighted Gerchberg–Saxton algorithm (CS-WGS).

## 3. Materials and Methods

### 3.1. Software Implementation

All algorithms described in Section 2 were implemented in Python 2.7 (Python Software Foundation, https://www.python.org/). All code is available as Supplementary Materials to this article, and in a public repository [17]. While better performances could be achieved implementing the algorithm in a lower level programming language, or through implementation over dedicated hardware (e.g., multi-core threads and GPU implementations), the Python scripting code was chosen to maximize the readability of the code, and provide an easy way to reproduce the results presented. Computational results reported in Section 4.1 were achieved with a mid-range desktop computer (Intel Core i5-4690 3.5 Ghz, 8 GB of memory, 64-bit Windows 7 operating system).

### 3.2. Experimental Setup

For the experimental investigation of the algorithms performances, we employed the system shown in Figure 1. two-photon excitation (TPE) beam (λ = 800 nm, diameter = 0.8 mm) from a Ti:Sapphire laser passes a beam expander composed by two lenses to uniformly illuminate the active area of a reflective SLM. The SLM is conjugated with the objective back aperture via lenses L1 and L2. An inverse pinhole blocks the zero-order beam while the CGH is projected onto the sample (Rhodamine 6 G uniformly coated slide, thickness 1.7 mm, Thorlabs; Newton, NJ, USA). Two-photon excitation fluorescence is collected by the same objective and, thanks to a short-pass dichroic mirror and a tube lens, is recorded by a complementary metal oxide semiconductor (CMOS) camera. To acquire images of three-dimensional holograms, a deformable mirror (DM-40, Thorlabs; Newton, NJ, USA) was added to the detection path, conjugated with the objective back aperture.

## 4. Results

### 4.1. Computational Results

The CS-GS algorithm was tested against the regular GS algorithm, as an upper limit of performance, and the random superposition algorithm, as an expected lower limit of performance. The performance metric used to evaluate the quality of the CGHs were, as in [12], the efficiency and uniformity of the obtained patterns. The efficiency *e* of the CGH is computed as the fraction of the total intensity directed at the generated spots. The uniformity *u* is calculated as
(8)$$u=1-\frac{I_{max}-I_{min}}{I_{max}+I_{min}},$$
where $I_{max}$ and $I_{min}$ are the intensities of the brightest and dimmest spot generated by the CGH.

As a computational test, CGHs were calculated for a random three-dimensional distribution of 100 spots, simulating a reasonable requirement for a optogenetics experiment, and for a grid of 10×10 spots in the image plane, representing a worst-case scenario for the uniformity of the pattern, where WGS should perform much better than GS. All variants of the GS algorithm were run for 50 iterations.

Due to the randomized initialization of the values of $\theta_{n}$, the final values of *u* and *e* of the CGH can vary slightly for different initializations. All the reported measurements were therefore performed by computing each CGH 10 times, with different random initializations, and calculating the mean and standard deviation of *u* and *e*.

The results, reported in Figure 2 and Figure 3, show how the CS-GS algorithm has performances virtually identical to GS with compression factors up to c=1/32, while providing an improvement in computational time of more than one order of magnitude. Moreover, CS-GS still greatly outperforms RS up to extreme compression factors.

As far as the weighted version is concerned, CS-WGS, only performing a single weighted iteration, does not perform as well as the regular version of WGS, especially in the case of the generation of regular geometrical patterns of spots. However, it still greatly outperforms GS in uniformity, and its deficit in uniformity compared to WGS can be considered practically negligible in the case of randomly distributed patterns of spots.

To rule out the possibility that the computation time difference between the standard and compressed sensing versions of the algorithms could be partially compensated by different convergence rates, efficiency and uniformity were estimated as a function of the iteration number. Since, for the considered experimental parameters, a compression factor of c=1/32 appears to represent the best compromise between speed and performances for both CS-GS and CS-WGS, such a compression parameter was used for the convergence speed comparison. As shown in Figure 2 and Figure 3, no significant difference in convergence speed can be observed. It is apparent from the convergence graphs that CS-GS and CS-WGS only differ from each other in the last iteration, as described in Section 2.3. This causes the sudden discontinuity in the CS-WGS convergence graph.

### 4.2. Experimental Results

To experimentally compare our methods to RS and iterative projections algorithms, we used each of them to compute the same 2D multi-spot CGH. This pattern contains 25 points arranged in letters shape for the initials of the laboratory institution (Technische Universiteit Delft, TUD) and extending over the complete available field of view. While all iteratively generated CGH were obtained with 100 iterations, we also operated WGS with 200 iterations and considered it as reference term in the performances evaluation. All the compressive sensing methods were computed with a sub-sampling value of 0.05.

As qualitatively shown in Figure 4, panels A–D CS-WGS and WGS outperform RS. The intensity profile from a line of points in each image (Figure 4, panel E) highlights the lack of excitation for some ROIs in the RS image (the peak corresponding to the third point is almost absent), while all other methods provide comparable results in terms of points intensity.

To quantitatively classify the performances of all algorithms including GS and CS-GS, we estimated the relative brightness and the intensity uniformity across the FOV of the different patterns. For each method, a stack of five images was acquired, and then, on the averaged image, seven random points were selected and fitted with 2D Gaussian profiles. The peak values from the fit are averaged and normalized for each method to produce the relative brightness. The brightness analysis (Figure 4, panel F) confirmed RS as the least efficient method, while all others show results comparable with the reference WGS.

From the standard deviation of the peaks values (Figure 4, panel G), we quantified the uniformity of the illumination. As anticipated in the computational results, WGS and CS-WGS algorithms are reaching uniformities that are very close to the golden standard (around 90%) at a fraction of the computational cost. Furthermore, we obtained a slightly higher uniformity for CS-WGS compared to WGS (run with the same number of iterations) corroborating that the deficit in uniformity found in the computational results is practically negligible for randomly distributed patterns.

Additionally, a separate dataset with spots distributed in three dimensions is reported in Figure 5, to prove the performances of the method in three dimensions. As can be observed, the WGS and CS-WGS images are practically indistinguishable, while RS, despite its minimal time saving compared to CS-WGS, has visibly worse performance, with significantly non uniform spots in the distribution.

## 5. Discussion

In this paper, we demonstrate, both computationally and experimentally, how the implementation of compressive sensing principles to the Gerchberg–Saxton algorithm for the generation of multi-spot holograms, in both its classic and weighted forms, can provide a dramatic decrease in computation time, without significantly affecting its performance. This method could therefore represent a standard in spatial light modulation based optogenetics stimulation, where fast and reliable generation of three-dimensional multi-spot patterns is key for a successful experiment. To facilitate implementation and testing for the neuroscience community, Python implementations of the RS, GS, WGS, CS-GS and CS-WGS algorithms are available as Supplementary Materials to this paper and in a public repository [17].

## Figures and Tables

**Figure 1 mps-02-00002-f001:**
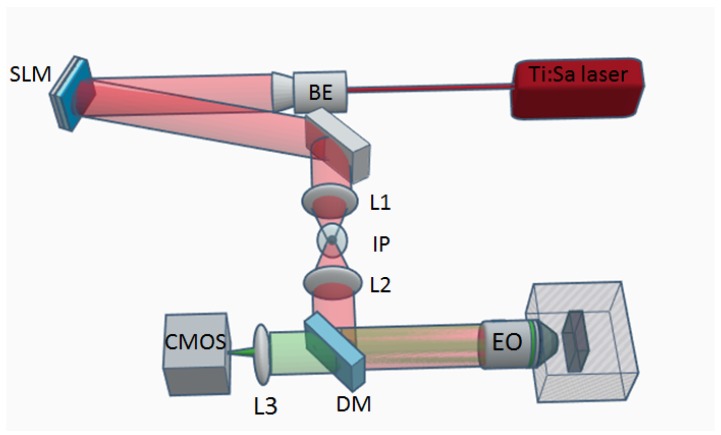
Scheme (not to scale) of two-photon excitation computer-generated hologram (CGH) setup. Ti:Sapphire laser (MIRA 900-F, Coherent; Santa Clara, CA, USA), beam expander (BE) composed by two lenses (f=25 mm and f=150 mm, respectively), reflective spatial light modulator (SLM, 1920 × 1152; pixel pitch, 9.2 μm Meadowlark Optics; Frederick, CO, USA), lens L1 (f=250 mm) and L2 (f=500 mm), inverse pinhole (IP), objective (EO, Olympus 20X, 1.0 NA, water immersion, Olympus; Tokyo, Japan), dichroic mirror (DM) with cutoff wavelength at 700 nm (FF01-7200/SDi01, Semrock; Rochester, NY, USA), tube lens L3 (f=200 mm), CMOS camera (IDS UI-3270CP, IDS; Obersulm, Germany).

**Figure 2 mps-02-00002-f002:**
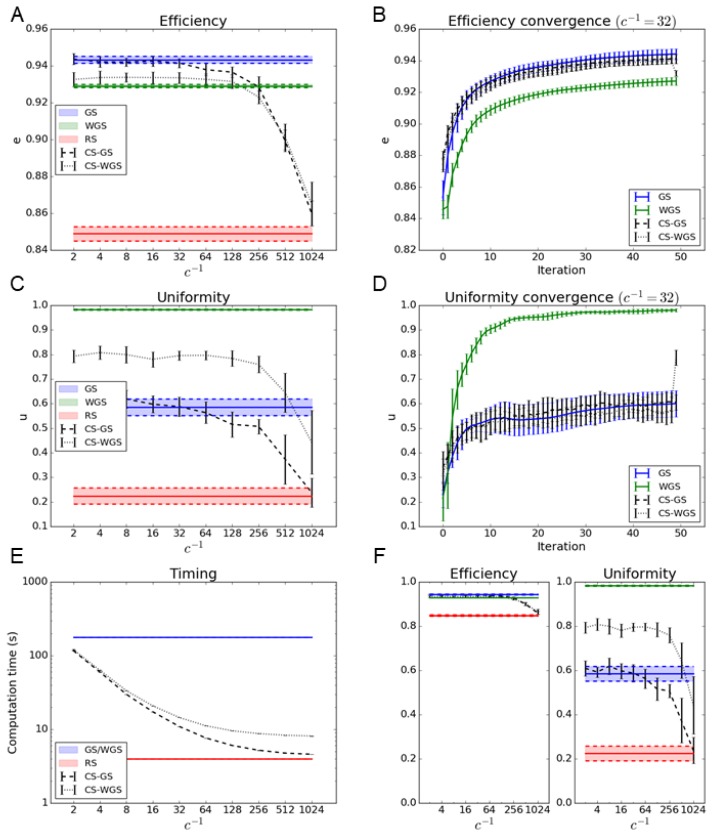
Algorithms performance in the case of the generation of a regular 2D array of 100 spots: (**A**,**B**) report the efficiency metric as a function of the subsampling parameter 1/c and the convergence graph at c=1/32 for the CS-GS and the CS-WGS algorithm. (**C**,**D**) report uniformity metric and convergence for the same dataset. For comparison, horizontal bars represent the interval of confidence of performance of GS, WGS, and RS algorithm. (**E**) reports the improvement in computation time as a function of 1/c. (**F**) reports the efficiency and uniformity represented in absolute scale to better represent overall performance.

**Figure 3 mps-02-00002-f003:**
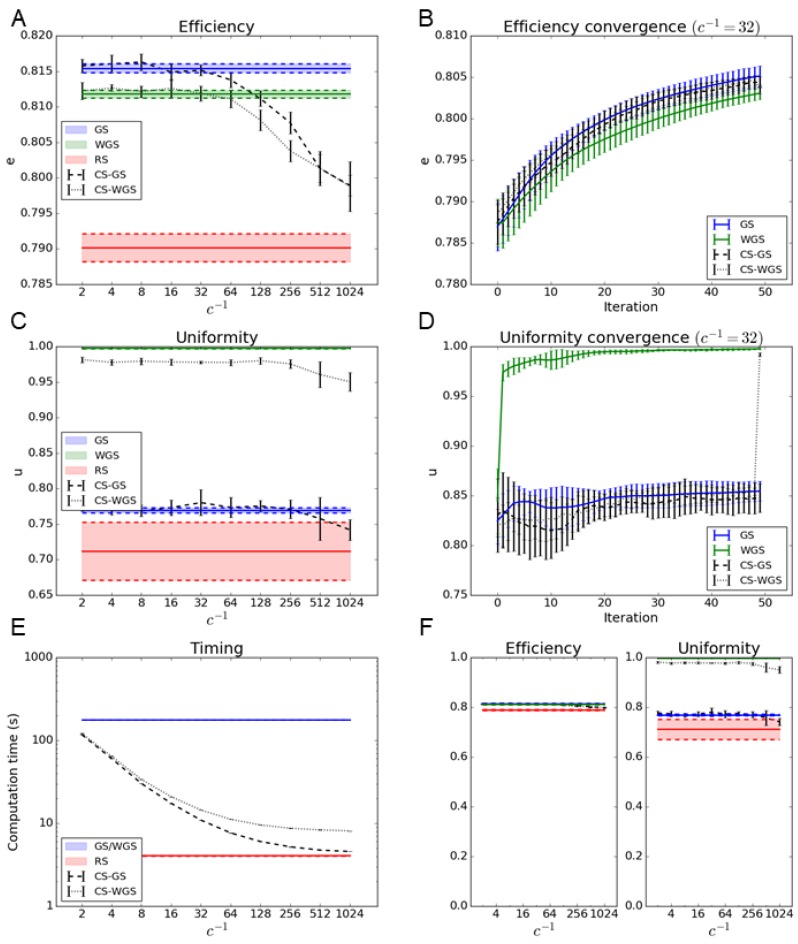
Algorithms performance in the case of the generation of a random 3D distribution of 100 spots: (**A**,**B**) report the efficiency metric as a function of the subsampling parameter 1/c and the convergence graph at c=1/32 for the CS-GS and the CS-WGS algorithm. (**C**,**D**) report uniformity metric and convergence for the same dataset. For comparison, horizontal bars represent the interval of confidence of performance of GS, WGS, and RS algorithm. (**E**) reports the improvement in computation time as a function of 1/c. (**F**) reports the efficiency and uniformity represented in absolute scale to better represent overall performance.

**Figure 4 mps-02-00002-f004:**
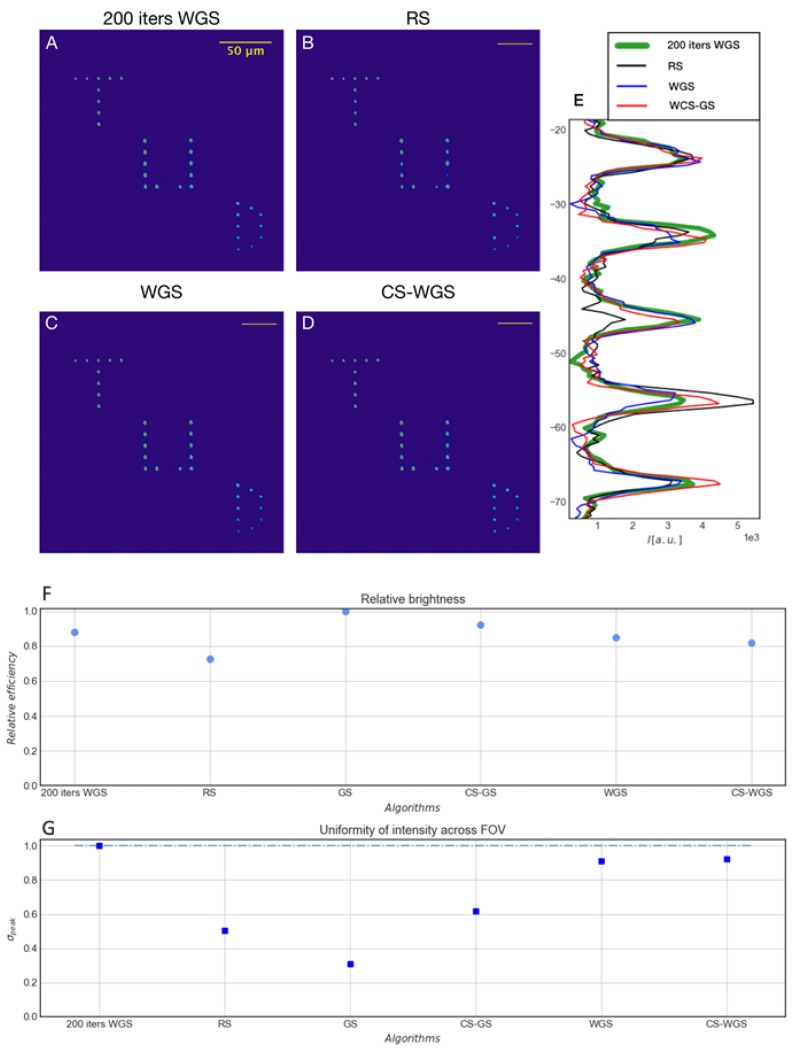
Experimental results: (**A**–**D**) Two photon excitation fluorescence images of multi-spot CGHs generated, respectively, with WGS 200 iterations, RS, WGS and CS-WGS algorithms; (**E**) intensity profile from a single line in panel (**D**); (**F**) relative brightness for all methods explored; and (**G**) uniformity across the field of view in the peak intensities for all methods explored.

**Figure 5 mps-02-00002-f005:**
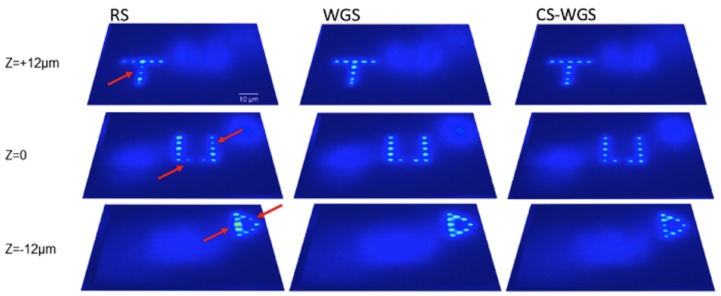
Experimental results for a three-dimensional hologram. CS-WGS and WGS were computed over 100 iterations, while CS-WGS had a compression factor of 0.05. Computation times were, respectively, 5 s, 40 s and 456 s for RS, WGS and CS-WGS. All images are reported on the same intensity scale. It can be noticed that, while the WGS and CS-WGS images are practically identical, RS has some spots with significantly different intensities (highlighted by the red arrows).

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
