# Peer review of "Fast Calculation of Computer Generated Holograms for 3D Photostimulation through Compressive-Sensing Gerchberg–Saxton Algorithm"

_mps, 2018, doi:10.3390/mps2010002_

Reviewer 1 Report

The authors propose an algorithm to calculate the CGH fast with the combination of compressive-sensing and GS algorithm. The authors numerically and experimentally verified the feasibility of the proposed method by qualitative and quantitative evaluations. The results show that the proposed algorithm has an advantage over other algorithms in calculation speed under the premise of guaranteeing image quality. However, the manuscript is lack of detail in some parts. The manuscript could be considered for publication, but not in its present form. Below is the comment.

1. There are too much background information in the abstract part. I think there should be more details about your work in this part for a quick understanding.

2. Figure 2 and Figure 3 are confusing for readers. I recommend the authors using lines of the same type and color for every algorithm in all graphs. Why is there a sudden change in CS-WGS algorithm of Figure D in both Figure 2 and Figure 3?

3. Your title of the manuscript includes ‘3D’, but the way to generate 3D image can’t be found throughout the manuscript, and 3D experiments are also not included in this manuscript. Therefore, the design process of 3D methods and experiments should be described in detail before publication.

Author Response

Response to Reviewer 1 Comments

Point 1: There are too much background information in the abstract part. I think there should be more details about your work in this part for a quick understanding.

Response 1: The abstract has been rewritten to address the comments of both reviewers. It now includes more details about the contents of the paper.

Point 2: Figure 2 and Figure 3 are confusing for readers. I recommend the authors using lines of the same type and color for every algorithm in all graphs.

Response 2: We agree the different colors for the convergence graphs would have been confusing for the reader. As suggested from the reviewer, we used the same colors for each algorithm in all graphs.

Point 3: Why is there a sudden change in CS-WGS algorithm of Figure D in both Figure 2 and Figure 3?

Response 3:As described in the method description, the CS-WGS algorithm only differs from CS-GS in the last iteration. This is now highlighted in the results section, where the following phrase has been added: "It is apparent, from the convergence graphs how CS-GS and CS-WGS only differ from each other in the last iteration, as described in subsection 2.3. This causes the sudden discontinuity in the CS-WGS convergence graph."

Point 4: Your title of the manuscript includes ‘3D’, but the way to generate 3D image can’t be found throughout the manuscript, and 3D experiments are also not included in this manuscript. Therefore, the design process of 3D methods and experiments should be described in detail before publication.

Response 4: The 3D nature of the algorithm is inherently included in the method, as the coordinates of all points generated with the algorithm are expressed in three dimensions. This should now be clearer in the introduction and abstract, due to the changes applied to address point 1, and the comments of reviewer 2.

 Moreover, to show more clearly the 3D performances of the method, we added an additional experimental dataset, showing the distribution of intensity for a hologram with spots distributed in the three dimensions.

Reviewer 2 Report

This work introduces a very important underexplored open area in the field of computer generated holography. While many other groups study CGH from an image rendering standpoint, this manuscript focuses on new applications, in particular applications of CGH to optogenetics in which point clouds are used to target and activate neurons on demand. The manuscript is clear and will benefit many groups currently relying on other slower CGH methods.   
I have two minor suggestions : - First, the authors should emphasize that their method is particularly adequate for "point cloud holography" where the goal is to create weighted distributions of rather sparsely distributed diffraction limited spots. Overuse of the word "Holography" suggests that the technique is valid for all types of holograms, which is not clear at this point. Although one can argue that a 3D object is a dense superposition of points, coherence may affect the quality of results in image rendering applications, also, the data in the manuscript only supports point distributions. Second, I would suggest referencing "Three-dimensional scanless holographic optogenetics with temporal focusing (3D-SHOT)." Nature Communications 8.1 (2017): 1228. ,or "Precise multimodal optical control of neural ensemble activity." Nat. Neurosci. 21 (2018), which discuss an existing application of holography for optogenetics that operates with point cloud holography, and would definitely benefit from the proposed algorithm presented in this manuscript. 

Author Response

Response to Reviewer 2 Comments

Point 1: The authors should emphasize that their method is particularly adequate for "point cloud holography" where the goal is to create weighted distributions of rather sparsely distributed diffraction limited spots.  Overuse of the word "Holography" suggests that the technique is valid for all types of holograms, which is not clear at this point. Although one can argue that a 3D object is a dense superposition of points, coherence may affect the quality of results in image rendering applications, also, the data in the manuscript only supports point distributions. Second, I would suggest referencing "Three-dimensional scanless holographic optogenetics with temporal focusing (3D-SHOT)." Nature Communications 8.1 (2017): 1228. ,or "Precise multimodal optical control of neural ensemble activity." Nat. Neurosci. 21  (2018), which discuss an existing application of holography for  optogenetics that operates with point cloud holography, and would  definitely benefit from the proposed algorithm presented in this  manuscript.

Response 1: We changed the abstract, introduction and discussion to better highlight the difference between image based CGH algorithms, which we do not consider in the paper, and multi-spot algorithms, which are the main focus of the paper.

Point 2: I would suggest referencing "Three-dimensional scanless holographic optogenetics with temporal focusing (3D-SHOT)." Nature Communications 8.1 (2017): 1228. ,or "Precise multimodal optical control of neural ensemble activity." Nat. Neurosci. 21  (2018), which discuss an existing application of holography for  optogenetics that operates with point cloud holography, and would  definitely benefit from the proposed algorithm presented in this  manuscript.

Response 2: In the introduction, we specified that multi-spot algorithms can be used to illuminate multiple extended regions when combined with scanning or temporal focusing, citing the suggested literature as an example of a temporal focusing application.
